# Enhanced cytotoxicity of T-DM1 in HER2-low carcinomas via autophagy inhibition

Jinghui Zhang[1☯], Xusheng Chang[2☯], Yingcheng Bai[1], Xiancai Ge[1], Kai Yin[2,*], Qun Xin[1*]

**1** Department of General Surgery, The 971st Hospital of Chinese People's Liberation Army Navy, Qingdao, P.R. China, **2** Department of Gastrointestinal Surgery, Changhai Hospital, Second Military Medical University, Shanghai, P. R. China.

☯ JZ and XC contributed equally to this paper as first author.
* xinqun@126.com (QX); kyin67@smmu.edu.cn (KY)

## Abstract

Ado-trastuzumab emtansine (T-DM1), a conjugate of trastuzumab and the cytotoxic agent emtansine, has demonstrated significant antitumor efficacy in HER2-positive (HER2+) carcinoma. However, its effectiveness is limited against carcinoma cells with low HER2 expression (HER2-low). Here, we demonstrate that targeting autophagy enhances the cytotoxicity of T-DM1 against HER2-low SGC7901 cells, highlighting the potential of autophagy modulation in improving T-DM1-based therapies for HER2-low carcinomas. Specifically, this study shows that T-DM1 exhibits limited cytotoxic effects on SGC7901 cells, but pharmacological inhibition of autophagy enhances its cytotoxicity. Moreover, transmission electron microscopy revealed that autophagy activation involved the three key phases of autophagic flux: the formation, fusion, and degradation of autophagosomes, while immunoblot analysis confirmed a reduction in Akt/mTOR signaling. Furthermore, autophagy inhibition accelerated the fusion of T-DM1 with lysosomes in SGC7901 cells, as shown by confocal microscopy. Collectively, these findings suggest that while T-DM1 alone induces limited cytotoxicity, combining it with autophagy inhibitors enhances its efficacy against HER2-low carcinoma cells. Mechanistically, autophagy inhibition increases the binding of T-DM1 to lysosomes, potentially facilitating the release of emtansine from the conjugate. These results present a novel strategy that combines T-DM1 with autophagy inhibitors to effectively treat HER2-low gastric cancer, thereby broadening the therapeutic scope of T-DM1 to encompass previously challenging cancer types.

## Introduction

Gastric cancer (GC) is a common gastrointestinal tumor receiving increasing attention due to its high incidence [1–3]. Human epidermal growth factor receptor 2 (HER2) is a crucial molecular target in precision oncology, but only approximately 15% of GC cases are identified as HER2-overexpressing [4–7]. While monoclonal

**Data availability statement:** All relevant data are within the paper and its Supporting Information files.

**Funding:** The author(s) received no specific funding for this work.

**Competing interests:** The authors have declared that no competing interests exist.

antibody drugs targeting HER2 are commonly used for HER2-positive (HER2+) carcinoma, their effectiveness in weakly HER2-positive (HER2-low) cancers has been limited [8].

Ado-trastuzumab emtansine (T-DM1) is a promising antibody-drug conjugate with potential for broad application. It enables the precise delivery of the tubulin inhibitor emtansine to HER2+tumor cells via the monoclonal antibody trastuzumab, which targets HER2 specifically. This leads to the targeted killing of HER2+tumor cells while sparing healthy cells [9,10]. Despite the significant clinical value in HER2+cancers, T-DM1 has shown limited therapeutic benefit for GC patients with low HER2 expression. These findings underscore the need to explore the mechanisms underlying the resistance of HER2-low GC to T-DM1, as this may uncover novel strategies to enhance the cytotoxicity of T-DM1 in these cases.

Autophagy is a critical cellular process that selectively degrades and recycles intracellular components, maintaining homeostasis in response to internal and external stress [11,12]. Autophagy can exhibit context-dependent effects in tumor therapy, and appropriate regulation of autophagy can enhance drug sensitivity [13]. While some studies suggest that activating autophagy can be cytotoxic and aid in tumor elimination, others indicate that autophagy may play a cytoprotective role, helping to sustain the aggressiveness and drug resistance of cancer cells [14–16]. For example, regulating autophagy with chloroquine has been shown to enhance antitumor efficacy in lung cancer [17]. Additionally, our previous studies have shown that T-DM1 can induce autophagy in HER2+GC, and that inhibiting this process can improve the therapeutic effect of T-DM1 [15]. However, the efficacy of T-DM1 and the role of autophagy in HER2-low GC are still unclear. Therefore, assessing autophagy in T-DM1-treated HER2-low cells may help elucidate the mechanism of T-DM1-mediated treatment and potentially expand the therapeutic range of T-DM1 in GC.

In our study, we investigated the cytotoxicity of T-DM1 on HER2-low SGC7901 cells. The results showed that T-DM1 triggered autophagy in SGC7901 cells. Notably, the combination of T-DM1 with the autophagy inhibitor LY294002 or 3-methyladenine (3-MA) markedly increased cytotoxic efficacy compared to T-DM1 alone. Mechanistically, we confirmed that T-DM1-induced autophagy was associated with the Akt/mTOR pathway. Additionally, inhibiting autophagy altered the phagocytosis and degradation efficiency of T-DM1 in lysosomes, facilitating the transport of emtansine from lysosomes to the cytoplasm in GC cells. In summary, these results highlight the cytoprotective effect of autophagy on SGC7901 cells treated with T-DM1, suggesting that combining T-DM1 with autophagy inhibitors could enhance treatment efficacy in HER2-low gastric cancer.

## Results

### T-DM1 exhibited limited cytotoxicity in HER2-low SGC7901 cells

We first compared HER2 expression levels across different carcinoma cell lines and confirmed that SGC7901 cells express a low level of HER2 compared to the HER2-negative MDA-MB-231 cells and the HER2-positive SK-BR-3 cells

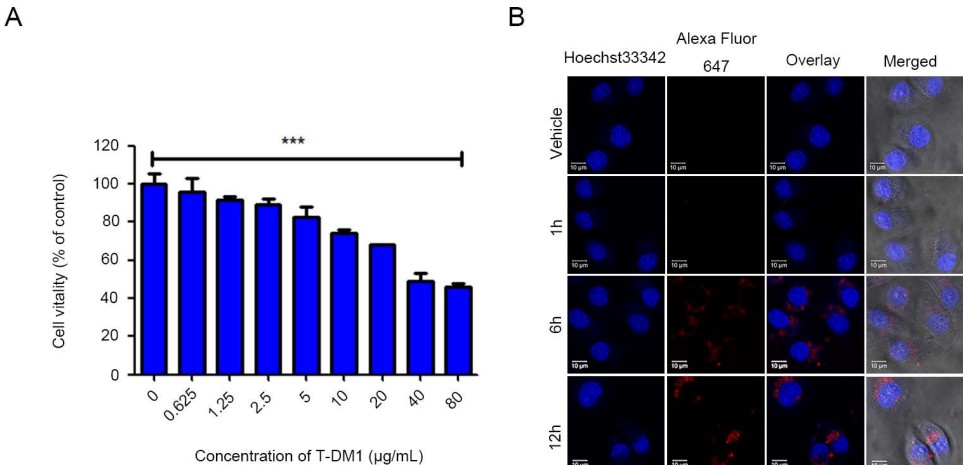

(Supplementary Figure S1). We next examined the cytotoxicity induced by T-DM1 in HER2-low SGC7901 cells using a Cell Counting Kit-8 (CCK-8) viability assay. It was shown that T-DM1 induced limited cytotoxicity and apoptosis in SGC7901 cells, indicated by a modest but significant reduction in cell viability even at high concentrations of T-DM1 (Fig 1A). We then tested the uptake process of T-DM1 in SGC7901 cells by labeling T-DM1 with red fluorescence. Laser confocal microscopy was used to track the uptake of T-DM1 at different time points, revealing that T-DM1 bound to the HER2 receptor at 1 hour and internalized at 12 hours post-treatment (Fig 1B).

Next, we assessed common cellular apoptotic markers to further confirm the cytotoxicity of SGC7901 cells induced by T-DM1. Flow cytometry (FCM) analysis of SGC7901 cells treated with T-DM1 at different concentrations indicated an increase in the Annexin V+ population, suggesting that T-DM1 enhanced the cytotoxicity of SGC7901 cells in a dose-dependent manner (Fig 2A). In addition, immunoblot analysis showed that cleaved caspase 3, cleaved caspase 9, and cleaved PARP were activated in a dose-dependent manner, indicating that T-DM1 induced apoptosis in SGC7901 cells (Fig 2B and C).

Together, these results suggest that T-DM1 can be taken up via endocytosis and induce limited cytotoxicity and apoptosis in HER2-low GC cells.

## Autophagosome formation and autophagic flux induced by T-DM1 in HER2-low SGC7901 cells

Three standard assays, including immunoblot analysis, confocal microscopy, and transmission electron microscopy (TEM), were employed to assess autophagy-related proteins and autophagosomes in SGC7901 cells following T-DM1 treatment. Immunoblot analysis showed that the expression of microtubule-associated protein 1 light chain 3 (LC3) was upregulated, while the expression of sequestosome 1 (SQSTM1, p62) was downregulated following T-DM1 treatment in SGC7901 cells (Fig 3A). TEM was used to track the accumulation and aggregation of double-membraned autophagic vesicles in SGC7901 cells in response to T-DM1 treatment (Fig 3B). Additionally, treatment with autophagy inducer rapamycin was used as a positive control to assess autophagy intensity. We observed that the number of green fluorescent puncta (autophagosomes) in SGC7901 cells treated with T-DM1 for 24 hours was comparable to that in the rapamycin-treated group (Supplementary Figure S2A–C), suggesting that T-DM1 induced autophagosome formation in SGC7901 cells.

**Fig 1. T-DM1 exhibited limited cytotoxicity and was taken up by SGC7901 cells. (A)** CCK-8 assays were used to evaluate the survival rates of SGC7901 cells treated with T-DM1 in a concentration-dependent manner for 72 hours (mean±**S**.D.; \*\*\*$P<0.001$, n=3). **(B)** Confocal microscopy revealed the optimal time point of T-DM1 fluorescence in SGC7901 cells, which was marked by the Alexa Fluor™ 647 Microscale Protein Labeling Kit.

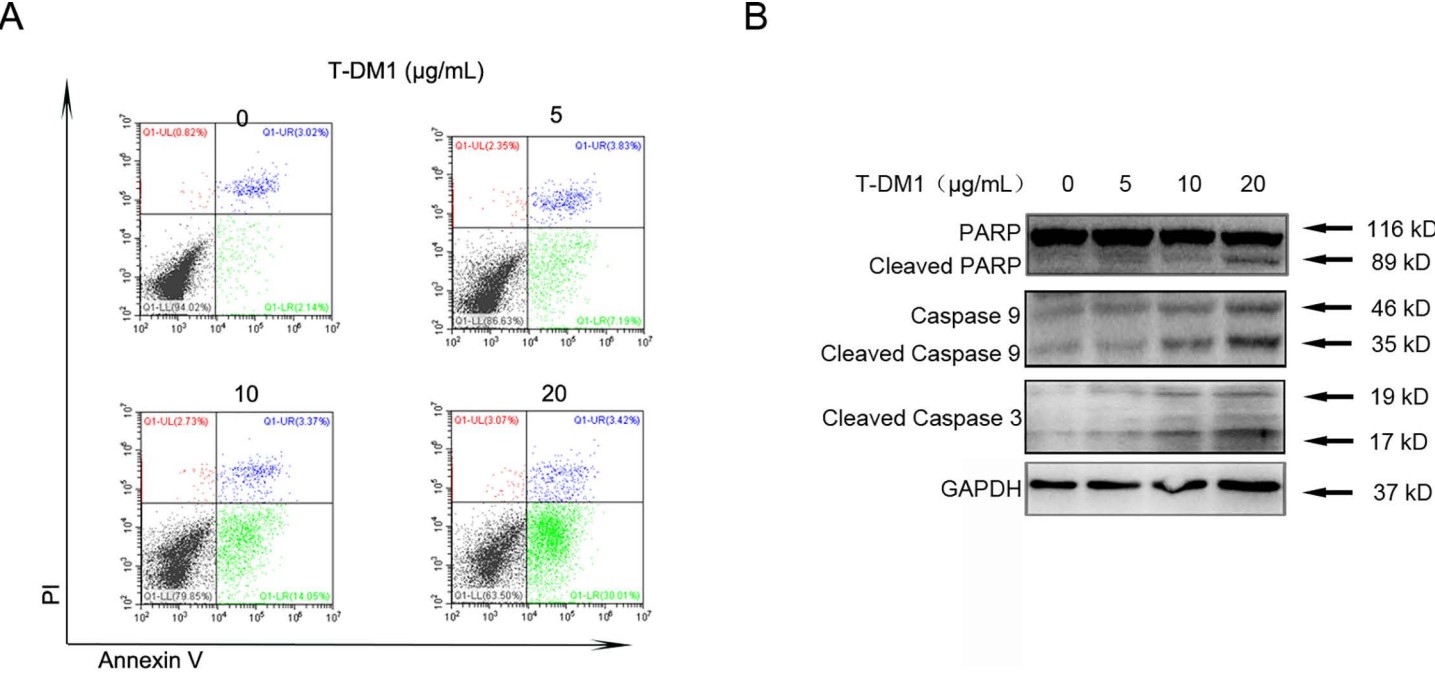

**Fig 2. T-DM1 dose-dependently induced apoptosis in SGC7901 cells. (A)** Apoptotic SGC7901 cells were stained with Annexin V-FITC/PI and observed after being incubated with T-DM1 for the indicated time (48 h) and were then examined by FCM. **(B)** Immunoblot analysis was used to measure the expression of apoptotic proteins (cleaved PARP, cleaved caspase 9, and cleaved caspase 3) after SGC7901 cells were treated with T-DM1 for 48 hours.

Moreover, confocal microscopy was used to monitor the three critical stages of autophagic flux in SGC7901 cells treated with T-DM1, visualized by staining with Cyto-ID and LysoTracker fluorescent dyes. The formation of autophagosomes was observed as early as 12 hours post-treatment, indicated by green fluorescence from Cyto-ID. Fusion of autophagosomes was detected at 24 hours post-treatment, indicated by yellow fluorescence. The degradation of autophagosomes was observed at 48 hours post-treatment, represented by orange fluorescence (Fig 3C).

These results indicate that T-DM1 can trigger the formation of autophagosomes and enhance autophagic flux in HER2-low SGC7901 cells.

### Autophagy inhibition enhanced T-DM1-mediated cytotoxicity and apoptosis in SGC7901 cells

To investigate the role of autophagy in SGC7901 cells in response to T-DM1 treatment, we used pharmacological inhibitors of autophagy, including LY294002 and 3-MA, to inhibit T-DM1-induced autophagy. We found that combining T-DM1 with 3-MA or LY294002 increased the expression level of SQSTM1 and decreased the expression level of LC3-II compared to T-DM1 alone (Fig 4A and B). Additionally, LY294002 or 3-MA enhanced T-DM1-induced cytotoxicity in SGC7901 cells (Fig 4E). Similarly, FCM results demonstrated a notable increase in the number of apoptotic cells when combining T-DM1 with LY294002 or 3-MA (Fig 4F). Immunoblotting also showed increased expression of apoptosis-related proteins (cleaved PARP and cleaved caspase 9) when T-DM1 was combined with LY294002 or 3-MA in SGC7901 cells compared to T-DM1 alone (Fig 4C and D; Supplementary Figure S3C and D).

In summary, these results indicate that cytoprotective autophagy can be markedly induced by T-DM1 in HER-low SGC7901 cells and that autophagy inhibition can enhance T-DM1-mediated cytotoxicity and apoptosis.

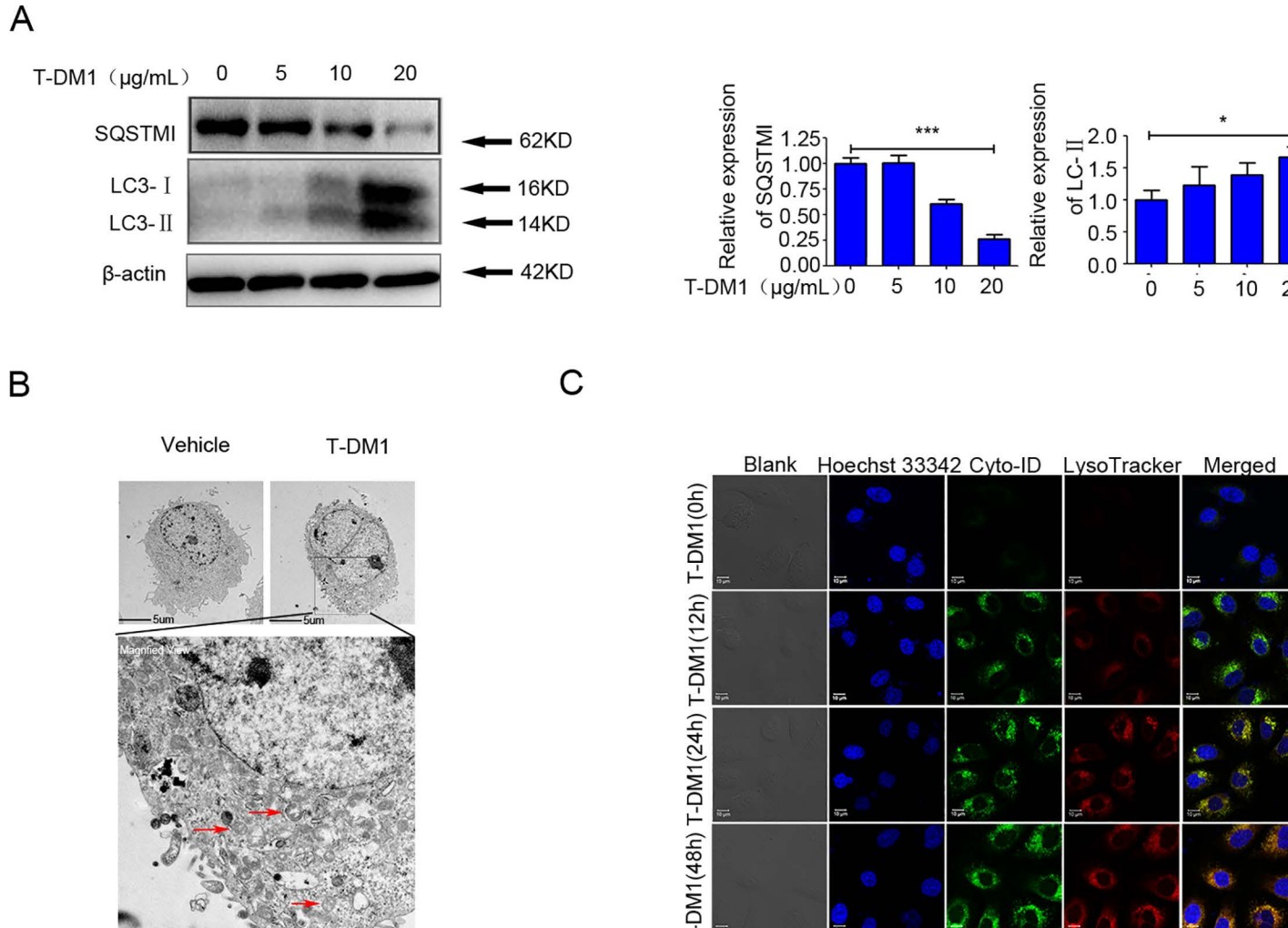

**Fig 3. Autophagosomes formation and autophagic flux were induced by T-DM1 in SGC7901 cells. (A)** Immunoblot analysis showed the expression of autophagic proteins (SQSTM1 and LC3-II) after SGC7901 cells were incubated with T-DM1 for 48 hours. The densitometric values were estimated by ImageJ software and normalized to the vehicle values. The data in the vehicle group were adjusted to 1.0, and data from three independent experiments are shown showed as the mean ± **S.**D. (Student's t-test; *$P$ < 0.05; ***$P$ < 0.001, n = 3). **(B)** Ultrastructural autophagosomes were examined by TEM and are displayed in an electron photomicrograph after SGC7901 cells were incubated with T-DM1 for 24 hours. **(C)** Autophagosomes were stained with Cyto-ID fluorescent dye (green) and lysosomes were stained with LysoTracker fluorescent dye (red) in SGC7901 cells incubated with T-DM1 and examined by confocal microscopy.

## The Akt/mTOR pathway is involved in the autophagy induction by T-DM1

To further investigate the molecular mechanisms underlying autophagy activation induced by T-DM1 in SGC7901 cells, immunoblot analysis was used to examine and quantify the expression of molecular components associated with the Akt/mTOR pathway and other essential downstream molecules. It was found that, alongside the autophagy induction by T-DM1, the phosphorylation of mTOR at S2448 was reduced in a dose-dependent manner. Additionally, the phosphorylation of Akt at S473, an upstream activator of mTOR, was also reduced in SGC7901 cells in response to T-DM1 (Fig 5A and B). Consistently, the expression of downstream components of the Akt/mTOR pathway, including 4E-BP1 and p70S6K, was also decreased by T-DM1 (Fig 5A and B).

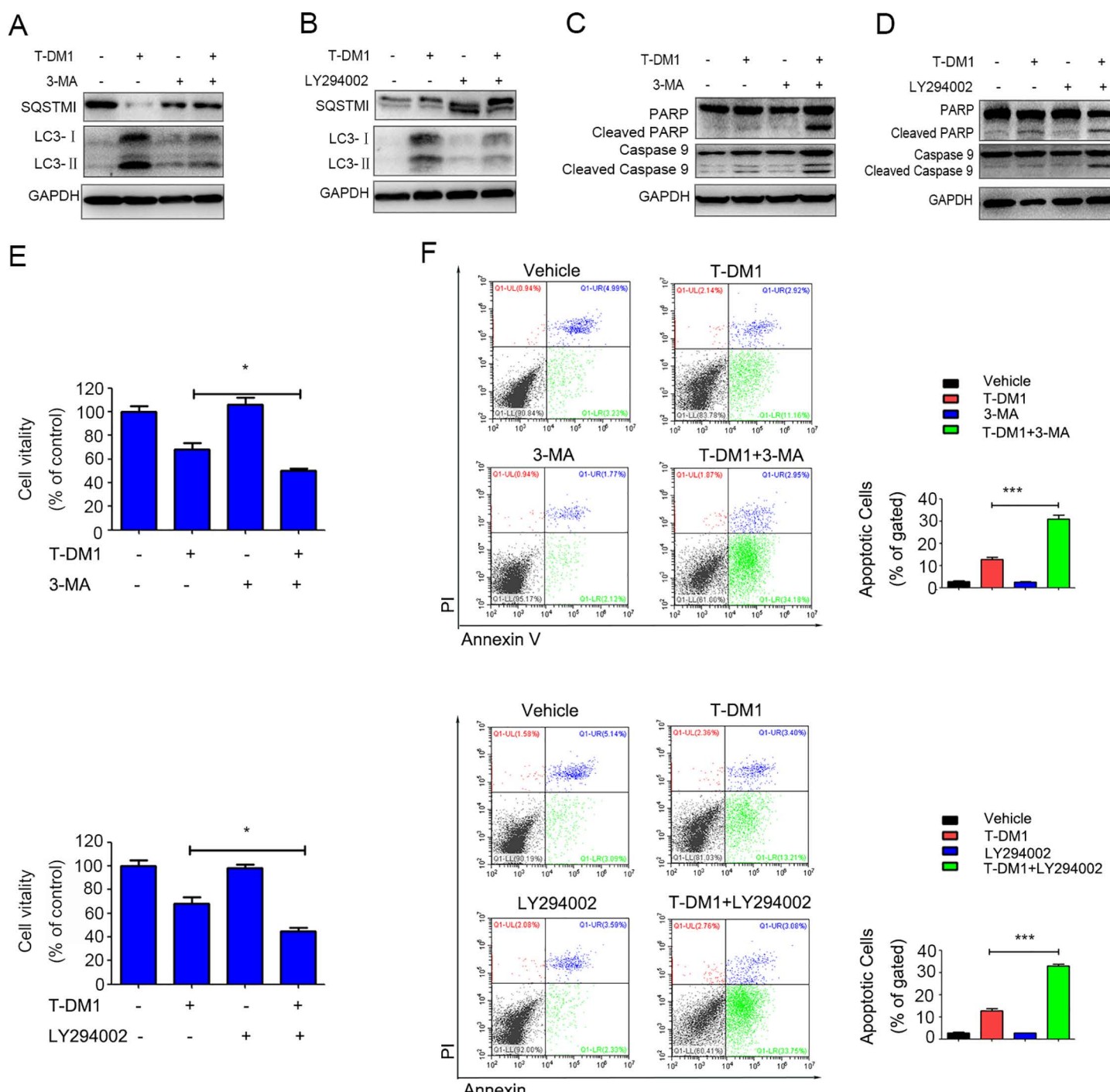

**Fig 4. Autophagy inhibition induced by T-DM1 enhanced the cytotoxicity of SGC7901 cells. (A-B)** Immunoblot analysis showed the expression levels of SQSTM1 and LC3-II in SGC7901 cells incubated with T-DM1 plus autophagy inhibitors for 48 hours. **(C-D)** Immunoblot analysis was used to determine the expression levels of cleaved PARP and cleaved caspase 9 in SGC7901 cells that were incubated with T-DM1 combined with auto-phagy inhibitors for 48 hours. **(E)** Autophagy inhibition with 3-MA or LY294002 reduced the viability of SGC7901 cells, as determined by relevant kits (mean ± S.D.; *$P < 0.05$ versus vehicle, n = 3). **(F)** The percentages of early apoptotic SGC7901 cells (the Annexin V + and PI- cells) were determined by FCM after incubation with 10 μg/ml T-DM1 and autophagy inhibitors (3-MA or LY294002). The results are presented in the side bar charts (***$P < 0.001$, n = 3).

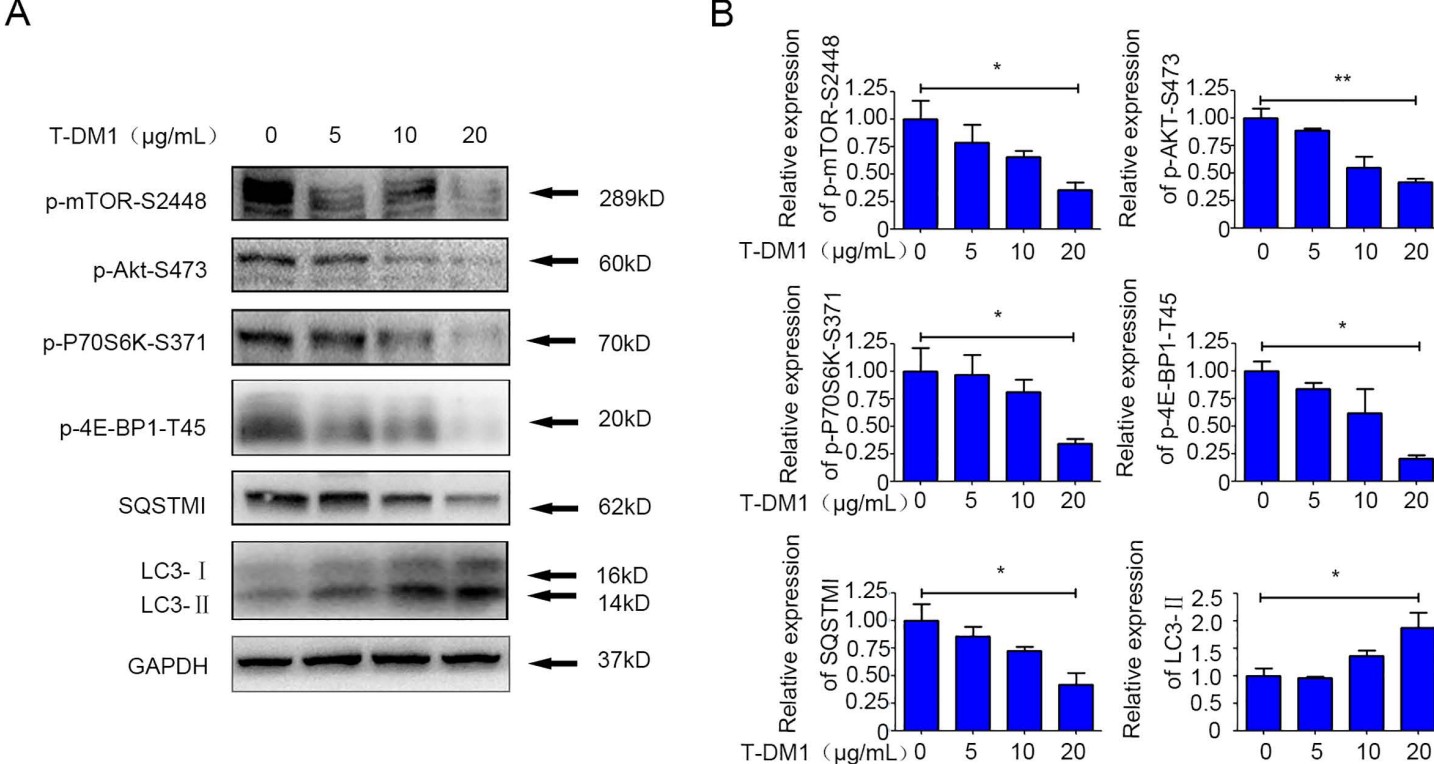

**Fig 5. The Akt/mTOR pathway and downstream components were involved in autophagy induced by T-DM1. (A)** SGC7901 cells were incubated with T-DM1 for 48 hours, and cell lysates were analyzed by immunoblotting to determine the expression levels of the Akt/mTOR pathway components. **(B)** The densitometric values of SGC7901 cell samples were evaluated using ImageJ software and normalized to the corresponding values of the vehicle group. The values of the vehicle group were adjusted to 1.0 (mean±S.D.; *P<0.05; **P<0.01 versus vehicle, n=3).

Together, these results suggest that the Akt/mTOR pathway is restrained in the induction of autophagy by T-DM1 in SGC7901 cells.

## Suppressing autophagy enhanced the phagocytosis and decomposition of T-DM1 by lysosomes

To further understand how autophagy suppression increased the cytotoxicity of T-DM1, we used confocal microscopy to track the transport of T-DM1 in SGC7901 cells. As reported previously, T-DM1 is internalized by cells and then releases the cytotoxic agent emtansine (DM1) in lysosomes [10]. Co-labeling of T-DM1 with Alexa Fluor 647 and of lysosomes with LysoTracker showed enhanced colocalization in SGC7901 cells when treated with both T-DM1 and LY294002, compared to treatment with T-DM1 or LY294002 alone. This suggests that autophagy inhibition accelerates the release of DM1 from lysosomes by increasing T-DM1 degradation (Fig 6).

Collectively, the fusion of T-DM1 with lysosomes was enhanced with autophagy inhibition, which facilitates the release emtansine from the conjugate and increases the cytotoxicity of T-DM1 in HER2-low SGC7901 cells.

## Discussion

GC, a prevalent malignant tumor, is receiving increasing attention because of its poor prognosis and high incidence [2]. HER2-targeted therapies are commonly used in clinical settings, but their application in HER2-low GC patients remains limited [4]. While HER2-targeted therapies have shown effectiveness in HER2+GC, patients with HER2-low GC exhibit

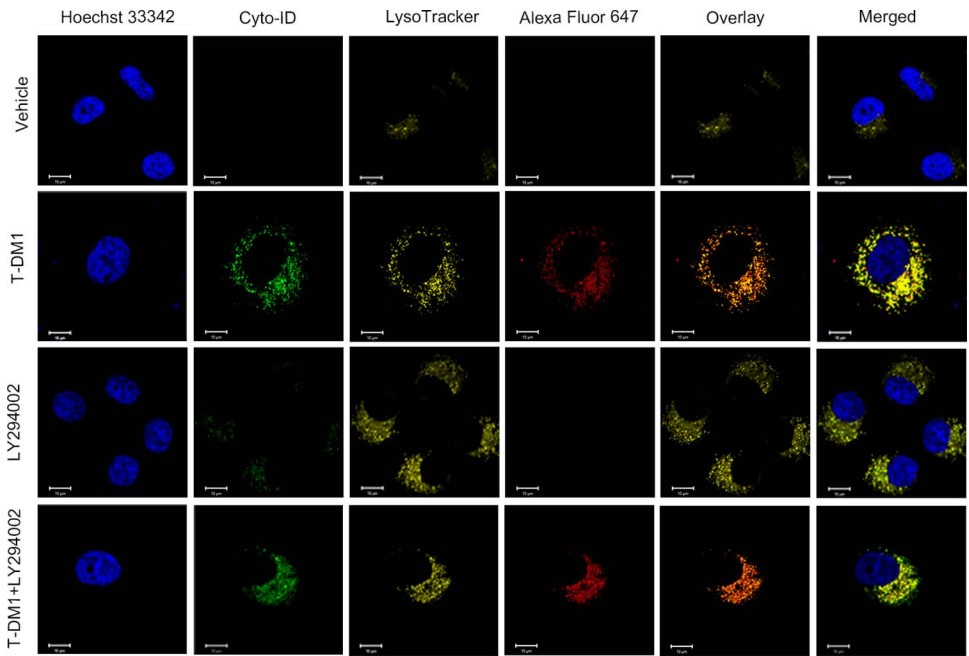

**Fig 6. Suppressing autophagy enhanced the fusion of T-DM1 with lysosomes.** T-DM1, labeled with Alexa Fluor™ 647, was visualized by red fluorescence. Lysosomes were stained with LysoTracker, which exhibited yellow fluorescence in SGC7901 cells. The cells were treated with T-DM1 and LY294002 for 24 hours and analyzed by confocal microscopy. Fusion was indicated by orange fluorescence, while autophagosomes were stained with Cyto-ID fluorescent dye and visualized by green fluorescence. The overlay of LysoTracker-stained lysosomes with Alexa Fluor™ 647-labeled T-DM1 showed the degree of fusion.

poor responses to these treatments [18–20]. Classification based on clinical features cannot fully describe biological characteristics, as nearly half of the HER2 negative BC tumors currently defined exhibit some degree of IHC expression, which has recently been renamed as HER2 low expression [21]. T-DM1, which is the first antibody-drug conjugate (ADC) approved for clinical use, is designed to decompose in lysosomes and disrupt microtubule organization, thereby selectively targeting HER2 + malignant tumor cells [20,22–24]. However, the effectiveness of T-DM1 is largely reduced in HER2-low tumors, so T-DM1 cannot be applied in clinical practice in HER2-low tumors. In recent years, people began to gradually pay attention to HER2-low gastric cancer, hoping to improve the effectiveness of targeted therapy [25]. This limited efficacy underscores the urgent need to better understand the mechanisms by which T-DM1 can be optimized for treating HER2-low GC.

This study aims to broaden the application of T-DM1 by investigating the role of autophagy in SGC7901 cells treated with T-DM1. Here, we examined the ultrastructure of autophagosomes, different stages of autophagic flux, and the expression of autophagy markers such as LC3-II, all of which confirmed the activation of autophagy in T-DM1-treated SGC7901 cells. However, the role of autophagy in T-DM1-treated SGC7901 cells remains unclear. The role of autophagy in cancer therapy remains under debate and is not yet systematically understood, especially in HER2-low GC [26–28]. Some studies have reported that autophagy inhibition creates metabolic vulnerability, leading to an energy crisis and apoptosis [29,30]. Conversely, others have indicated that activating autophagy can enhance the killing of cancer cells under hypoxic conditions [31]. Our results demonstrate that autophagy inhibition enhances the cytotoxicity of T-DM1, suggesting that autophagy activated by T-DM1 has a cytoprotective effect on SGC7901 cells, thereby limiting the efficacy of T-DM1.

The Akt/mTOR pathway plays a critical role in regulating autophagy and drug sensitivity in eukaryotic cells [32,33]. Our results also indicated that inactivation of the Akt/mTOR signaling pathway and its downstream molecules was associated

with the autophagy activation induced by T-DM1. Moreover, autophagy inhibition enhanced the fusion of T-DM1 with lysosomes, facilitating the release of emtansine from the conjugate. Therefore, we conclude that autophagy blocks the lysosomal degradation of T-DM1 and prevents the release of emtansine from the conjugate. This mechanism helps explain how T-DM1's cytotoxicity is enhanced when combined with autophagy inhibitors, and complements existing studies on T-DM1 resistance related to defects in lysosomal functions [10,34,35].

ADCs represent a promising class of chemotherapy drugs that are engineered to deliver cytotoxic agents directly to target cells via monoclonal antibodies [36]. Ideally, target cells should remain highly sensitive to ADCs. However, our study showed that T-DM1 did not exhibit a superior therapeutic advantage against HER2-low GC cells [37]. Novel ADC drugs have been tested in clinical trials in HER2-low tumors, which suggests that fewer available HER2 receptors on cancer cells may be sufficient to achieve clinical benefits [38]. We observed the processing of T-DM1 from receptor recognition to intracellular lysosomes in SGC7901 cells, finding that fusion with lysosomes was strengthened by autophagy inhibition. Increased fusion of T-DM1 with lysosomes effectively increased the cytotoxic effect of T-DM1 on SGC7901 cells. Importantly, our findings, along with previous studies, suggest that autophagy interference in the dissociation of ADCs may be a universal phenomenon across different tumors [15,39]. Therefore, regulating autophagy could potentially alter the sensitivity of ADC-based cancer treatment and could be a viable strategy to broaden the therapeutic range of ADCs.

In summary, T-DM1 induced cytoprotective autophagy in SGC7901 cells, and combining T-DM1 with autophagy inhibitors enhanced its cytotoxicity. Unlike developing novel ADCs targeting HER2-low tumors, our results suggest a novel strategy for improving the treatment of HER2-low GC by using ADCs in conjunction with autophagy inhibitors.

## Materials and Methods

### Cell line and culture

The SGC7901, N-87, MDA-MB-231 and SKBR-3 cell lines was obtained from Minhang Hospital of Fudan University and grown in Roswell Park Memorial Institute (RPMI) 1640 medium supplemented with penicillin, streptomycin, and fetal bovine serum in a cell incubator with constant temperature and humidity.

### Reagents and antibodies

The reagents LY294002 (Code: S1105) and 3-MA (Code: S2767) were purchased from Selleck (Shanghai, China). T-DM1 was obtained from Roche, and Cyto-ID fluorescent dye (Code: ENZ-51031-K200) was purchased from Enzo Life Sciences. The following primary antibodies, including phospho-mTOR (Ser2448), phospho-Akt (Ser473), phospho-70S6 kinase (Ser371), phospho-4E-BP1/2/3 (Thr46), caspase 3, caspase 9, PARP, LC3, SQSTM1, CycinB1 and HER2, were provided from Cell Signaling Technology for immunoblot analysis. In addition, horseradish peroxidase-conjugated goat anti-rabbit immunoglobulin G Secondary antibodies were provided by MR Biotech, and other reagents were purchased from Beyotime (Haimen, China).

### Cell viability assay

Cell Counting Kit-8 (CCK-8) was used to determine the viability of SGC7901 cells. Approximately 5,000 cells were inoculated per well. Then, the cells were treated with T-DM1 or autophagy inhibitors for certain times. CCK-8 was added immediately in the dark and incubated for 1.5h [40]. Finally, the absorbance was analyzed at a specific wavelength (450nm) by spectrometry.

### Immunoblot analysis

To analyze the expression level of the target proteins, total proteins were collected from SGC7901 cells by subjecting them to cell lysis buffer for 0.5h at 0 °C. We centrifuged the lysates for 10min and quantified the proteins with a BCA kit. We

separated same amounts of protein by SDS-PAGE according to molecular weight and transferred them to PVDF membranes under constant current conditions. Then, the PVDF membrane containing the target proteins was incubated with the corresponding antibodies. Ultimately, chemiluminescence reagents were used to visualize the immunoreactive bands, and ImageJ software (National Institutes of Health, USA) was used to compare the intensities of the immunoreactive bands.

## Transmission electron microscopy

SGC7901 cells were treated with T-DM1 for the indicated times (24 h) and then subjected to the experimental process as described [41]. The sections were visualized by a JEM 1410 transmission electron microscope (JEOL, Inc.) according to the instructions.

## Confocal microscopy

SGC7901 cells and T-DM1 were cultured for the indicated times. The cell samples were stained with LysoTracker, Hoechst 33342 and Cyto-ID fluorescent dye according to the instructions [42]. Furthermore, 50 nM rapamycin was used as a positive control. An LSM710 confocal microscope (Carl Zeiss, Germany) was used to analyze all samples. In particular, the concentration of T-DM1 was 10 µg/ml.

## T-DM1 phagocytosis assay

SGC7901 cells were cultured with T-DM1, which marked with the Alexa Fluor™ 647 Microscale Protein Labeling Kit (Invitrogen, USA), to analyze the phagocytic activity after different times at 37 °C. Confocal microscopy was used to observe the cell samples.

## Apoptosis analysis

Apoptotic SGC7901 cells were analyzed by apoptosis assay kits (Annexin V-FITC/PI) which were purchased from BD Bioscience (USA). Each sample was collected, washed, and stained for 15 min at 37 °C. Approximately 10,000 cells were examined in all samples by flow cytometry (Fullerton, CA, USA).

## Confocal immunofluorescence analysis

SGC7901 cells were incubated in microscopy chambers, fixed with 4% paraformaldehyde (10 min), permeabilized with 0.1% polyethylene glycol octylphenol ether (15 min), and blocked with 10% bovine serum albumin (120 min). The samples were washed with phosphate-buffered saline and labeled with PKH26 after being incubated with anti-HER2 antibodies at 4 °C (12 h). Finally, the cells were observed under a confocal microscope after being stained with Hoechst 33342 and installed on dishes.

## Statistical analysis

GraphPad Prism 8 (San Diego, CA) was used to analyze the data. The results were analyzed by Student's t-test and are presented as the mean ± S.D. Differences were believed statistically significant when $P < 0.05$.

## Supplementary Material

**Supplementary Figure S1.** **(A)** The expression of HER2 was examined by immunofluorescence in SGC7901, HER2-negative MDA-MB-231 and HER2 + SK-BR-3 cells using a confocal microscope. **(B)** The brightness values of HER2 red dye staining were estimated using ImageJ software and the densitometric values were normalized to the corresponding values of the vehicle group. The values of the vehicle group were set as 1.0 (n = 3 independent experiments; mean ± S.D.; **$P < 0.01$ and ***$P < 0.001$). **(C)** CCK-8 assays were used to evaluate the survival rates of SK-BR-3 cells

(HER2-high Breast cancer cells) treated with T-DM1 in a concentration-dependent manner for 72 hours (mean±S.D.; ***P<0.001, n=3).
(TIF)

**Supplementary Figure S2.** **(A)** Confocal immunofluorescence was used to verify autophagosome staining with Cyto-ID fluorescent dye in SGC7901 cells after treatment with T-DM1 for 24 hours, with the group incubated with rapamycin serving as the control. **(B)** The brightness values of Cyto-ID green dye staining were estimated using ImageJ software and the densitometric values were normalized to the corresponding values of the vehicle group. The values of the vehicle group were set as 1.0 (n=3 independent experiments; mean±S.D.; ***P<0.001). **(C)** The brightness values of the Alexa Fluor™ 647 were evaluated using ImageJ software, and the values were normalized to the corresponding values of the vehicle group. **(D)** The brightness values of autophagic flux were evaluated using ImageJ software, and the values were normalized to the corresponding values of the vehicle group. The values of the vehicle group were set as 1.0 (n=3 independent experiments; mean±S.D.; **P<0.01 and ***P<0.001).
(TIF)

**Supplementary Figure S3.** **(A)** Apoptotic cells (the Annexin V+ and PI- cells) examined by FCM for the indicated time (48h) were counted and analyzed, with results shown in bar charts (n=3; mean±S.D.; ***P<0.001). **(B-D)** Densitometric values of the expression of apoptotic proteins (cleaved PARP, cleaved caspase 9, and cleaved caspase 3) after SGC7901 cells treated with T-DM1 for 48 hours were calculated using ImageJ software and normalized to the value of the corresponding vehicle bands. The values of the vehicle group were adjusted to 1.0 and are shown as the mean±S.D. Student's t-test was used to analyze the data. *P<0.05 and ***P<0.001, n=3.
(TIF)

**Supplementary Figure S4.** **(A-D)** Densitometric values of SQSTM1, LC3-II, cleaved PARP, and cleaved caspase 9 in SGC7901 cells treated with T-DM1 and autophagy inhibitor (3-MA or LY294002) were analyzed using ImageJ software (n=3; mean±S.D.; *P<0.05, **P<0.01, and ***P<0.001).
(TIF)

**Supplementary Figure S5.** The brightness values of the Alexa Fluor™ 647 visualized by red fluorescence, Cyto-ID visualized by fluorescence and LysoTracker exhibited yellow fluorescence were evaluated using ImageJ software, and the values were normalized to the corresponding values of the vehicle group.
(TIF)

**Supplementary Figure S6.** SGC7901 cells were incubated with T-DM1 and 3-MA for 48 hours, and cell lysates were analyzed by immunoblotting to determine the expression levels of the CycinB1. The densitometric values of SGC7901 cell samples were evaluated using ImageJ software and normalized to the corresponding values of the vehicle group. The values of the vehicle group were adjusted to 1.0 (mean±S.D.; *P<0.05; **P<0.01 versus vehicle, n=3).
(TIF)

**Supplementary Figure S7.** The cell lysates of MDA-MB-231 (HER2-negative Breast cancer cell line), N-87 cells (HER2-positive GC cancer cell line), SGC7901 and SK-BR-3 cells (HER2-positive Breast cancer cell line) were analyzed by immunoblotting to determine the expression levels of the HER2.
(TIF)

## Supporting Information

**S1 raw images:** A supporting information caption for the file 'S1_raw_images.pdf.
(PDF)

## Author contributions

**Conceptualization:** Qun Xin, Kai Yin.

**Data curation:** Jinghui Zhang.

**Formal analysis:** Xusheng Chang, Yingcheng Bai.

**Investigation:** Xiancai Ge.

**Methodology:** Yingcheng Bai.

**Resources:** Qun Xin, Kai Yin.

**Writing – original draft:** Jinghui Zhang, Xusheng Chang.

**Writing – review & editing:** Qun Xin, Kai Yin.

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
