## [Decision Letter · Decision Letter 0]

15 Dec 2024

PONE-D-24-51611Enhanced Cytotoxicity of T-DM1 in HER2-low Carcinomas via Autophagy InhibitionPLOS ONE

Dear Dr. Xin,

Thank you for submitting your manuscript to PLOS ONE. After careful consideration, we feel that it has merit but does not fully meet PLOS ONE’s publication criteria as it currently stands. Therefore, we invite you to submit a revised version of the manuscript that addresses the points raised during the review process.

We look forward to receiving your revised manuscript.

Kind regards,

Ruo Wang

Academic Editor

PLOS ONE

Journal Requirements: When submitting your revision, we need you to address these additional requirements. 1. Please ensure that your manuscript meets PLOS ONE's style requirements, including those for file naming. The PLOS ONE style templates can be found at https://journals.plos.org/plosone/s/file?id=wjVg/PLOSOne_formatting_sample_main_body.pdf and https://journals.plos.org/plosone/s/file?id=ba62/PLOSOne_formatting_sample_title_authors_affiliations.pdf 2. When completing the data availability statement of the submission form, you indicated that you will make your data available on acceptance. We strongly recommend all authors decide on a data sharing plan before acceptance, as the process can be lengthy and hold up publication timelines. Please note that, though access restrictions are acceptable now, your entire data will need to be made freely accessible if your manuscript is accepted for publication. This policy applies to all data except where public deposition would breach compliance with the protocol approved by your research ethics board. If you are unable to adhere to our open data policy, please kindly revise your statement to explain your reasoning and we will seek the editor's input on an exemption. Please be assured that, once you have provided your new statement, the assessment of your exemption will not hold up the peer review process. 3. PLOS requires an ORCID iD for the corresponding author in Editorial Manager on papers submitted after December 6th, 2016. Please ensure that you have an ORCID iD and that it is validated in Editorial Manager. To do this, go to ‘Update my Information’ (in the upper left-hand corner of the main menu), and click on the Fetch/Validate link next to the ORCID field. This will take you to the ORCID site and allow you to create a new iD or authenticate a pre-existing iD in Editorial Manager. 4. PLOS ONE now requires that authors provide the original uncropped and unadjusted images underlying all blot or gel results reported in a submission’s figures or Supporting Information files. This policy and the journal’s other requirements for blot/gel reporting and figure preparation are described in detail at https://journals.plos.org/plosone/s/figures#loc-blot-and-gel-reporting-requirements and https://journals.plos.org/plosone/s/figures#loc-preparing-figures-from-image-files. When you submit your revised manuscript, please ensure that your figures adhere fully to these guidelines and provide the original underlying images for all blot or gel data reported in your submission. See the following link for instructions on providing the original image data: https://journals.plos.org/plosone/s/figures#loc-original-images-for-blots-and-gels.   In your cover letter, please note whether your blot/gel image data are in Supporting Information or posted at a public data repository, provide the repository URL if relevant, and provide specific details as to which raw blot/gel images, if any, are not available. Email us at plosone@plos.org if you have any questions. 5. We notice that your supplementary figures are uploaded with the file type 'Figure'. Please amend the file type to 'Supporting Information'. Please ensure that each Supporting Information file has a legend listed in the manuscript after the references list.

Reviewers' comments:

Reviewer's Responses to Questions

**Comments to the Author**

1. Is the manuscript technically sound, and do the data support the conclusions?

Reviewer #1: Partly

Reviewer #2: Yes

Reviewer #3: Partly

2. Has the statistical analysis been performed appropriately and rigorously? 

Reviewer #1: Yes

Reviewer #2: Yes

Reviewer #3: No

3. Have the authors made all data underlying the findings in their manuscript fully available?

Reviewer #1: Yes

Reviewer #2: Yes

Reviewer #3: Yes

4. Is the manuscript presented in an intelligible fashion and written in standard English?

Reviewer #1: Yes

Reviewer #2: Yes

Reviewer #3: Yes

5. Review Comments to the Author

Reviewer #1: The manuscript explores the potential of combining T-DM1, an antibody-drug conjugate, with autophagy inhibitors to enhance its cytotoxicity in HER2-low gastric carcinoma cells. The authors concluded that T-DM1 alone exhibits limited cytotoxicity in HER2-low carcinoma cells due to the activation of cytoprotective autophagy. However, inhibiting autophagy pharmacologically significantly increases T-DM1’s cytotoxic effects by promoting its lysosomal fusion and facilitating the release of its active cytotoxic agent, emtansine.

Comments:

The study largely repurposes previously explored concepts by combining T-DM1 with autophagy inhibitors, an approach that has already been examined in other contexts. It does not provide groundbreaking insights or novel mechanisms specific to HER2-low gastric cancer.

While the manuscript mentions the involvement of the Akt/mTOR pathway and autophagy modulation, it fails to thoroughly investigate the molecular interactions or provide sufficient evidence to establish causal relationships between autophagy inhibition and the enhanced efficacy of T-DM1.

The figures and results sections include excessive repetition, particularly in the presentation of apoptotic and autophagy-related assays. Streamlining this information would improve clarity and enhance the manuscript's readability.

The manuscript lacks detailed descriptions of the statistical analyses performed, including criteria for significance and the sample sizes used for each experiment. This omission undermines the reproducibility and robustness of the findings.

The study does not adequately compare the efficacy of the proposed combination therapy with existing treatment options for HER2-low gastric cancer, limiting its clinical relevance and applicability.

Phrases such as "significantly enhanced cytotoxicity" are used without providing clear quantitative benchmarks, leaving the practical significance of the findings unclear.

The discussion fails to place the findings within the broader landscape of HER2-low gastric cancer research. Conflicting evidence or alternative interpretations are not sufficiently addressed, leaving gaps in the contextual analysis.

The authors assert that autophagy inhibition may have universal applications across various tumor types but provide no supporting evidence beyond the context of HER2-low gastric cancer cells. This weakens the broader impact of the conclusions.

Reviewer #2: In the manuscript entitled: “Enhanced cytotoxicity of T-MD1 in HER2-low carcinomas via autophagy inhibition”, Zhang et al. showed that T-DM1 induces autophagy and combination of T-DM1 with autophagy inhibitors increases T-DM1 efficacy in HER2-low gastric cancer. A potential expansion of the use of T-DM1 and may be other HER2-targeted therapies to tumors with low HER2 expression is significant since the HER2-low patient subpopulation usually exhibits an aggressive disease progression. Despite that, there are several major points that need to be addressed as listed below:

Major points:

1- All the IF images need quantification.

2- In Fig. 1B, it was claimed that endocytosis happens at 6 hrs, however, it is not visible from the image provided.

3- Is there any change in T-DM1 internalization upon combination of T-DM1 with autophagy inhibitors.

4- It is not clear how the apoptotic cells were quantified. Were they the Annexin V+ and PI- cells corresponding to early apoptosis or Annexin V+ PI+ cells corresponding to late apoptosis?

5- Is DM1 treatment also inducing autophagy induction upon mTOR inhibition?

6- The reduction of cell viability upon combination of T-DM1 with the autophagy inhibitors is not convincing.

7- Since the major mechanism of action of T-DM1 is mitotic arrest, it is suggested to examine the mitotic arrest markers in HER2-low cells treated with T-DM1 and autophagy inhibitor combination.

8- To eliminate the potential cell line-specific effects, it is suggested that the key experiments, e.g., reduced cell viability and apoptosis induction upon combination of T-DM1 with the autophagy inhibitors are repeated in a second HER2-low gastric cancer cell line.

Reviewer #3: In this manuscript, Zhang and colleagues aimed to enhance the therapeutic efficacy of Ado-trastuzumab emtansine (T-DM1) in HER2-low gastric carcinoma cells, which typically exhibit limited responsiveness to this treatment. By investigating the role of autophagy in T-DM1 resistance, the authors discovered that T-DM1 induces cytoprotective autophagy via the Akt/mTOR pathway in HER2-low SGC7901 cells. This autophagy activation mitigates the drug's cytotoxic effects. However, pharmacological inhibition of autophagy significantly enhanced T-DM1-induced cytotoxicity and apoptosis by promoting lysosomal degradation of the drug and facilitating the release of its cytotoxic payload, emtansine. These findings highlight a novel strategy for improving T-DM1 efficacy in HER2-low gastric cancer by combining it with autophagy inhibitors, potentially broadening the therapeutic applications of T-DM1.

That said, there are several points that should be addressed before the manuscript is suitable for publication:

Major Comments:

1. In Figure 1A, the authors demonstrate that the HER2-low cell line exhibits limited responsiveness to T-DM1. To strengthen this claim, it would be beneficial to include a HER2-positive cell line as a control. A direct comparison of T-DM1 cytotoxicity between HER2-high and HER2-low cell lines would provide clearer evidence of the differential sensitivity.

2. The manuscript discusses the Akt/mTOR pathway's involvement in autophagy induced by T-DM1. However, to provide stronger mechanistic insight, the authors should consider using specific inhibitors of this pathway. This would validate whether the Akt/mTOR pathway is indeed a critical mediator of autophagy following T-DM1 treatment, as suggested in Figure 5.

3. While statistical analyses are mentioned, the manuscript lacks consistent information on the specific tests used for each figure. Including detailed descriptions of the statistical tests applied and the rationale for their selection would enhance the credibility and reproducibility of the findings.

4. The discussion could benefit from a more comprehensive comparison with existing studies on T-DM1 and other ADCs in HER2-low cancers, such as breast cancer. Additionally, elaborating on the broader therapeutic implications, such as the potential application of autophagy inhibitors in other HER2-low cancers or ADC therapies, would add significant value.

Minor Comments:

- The WB analysis for HER2 expression in Supplemental Figure 1 should be prioritized over immunofluorescence (IF) for its quantitative accuracy. Including a HER2-positive GC cancer cell line in the comparison panel would further enhance the relevance of the findings.

- There is some inconsistency in the use of terms such as "autophagy inhibition" and "blocking autophagy." Ensuring consistency in terminology throughout the manuscript would improve readability. A professional language review might also be beneficial to refine clarity and conciseness.

By addressing these points, the manuscript will be significantly strengthened and better positioned for publication. The study presents important insights into enhancing T-DM1 efficacy in HER2-low gastric cancer, and these revisions would further solidify the impact and rigor of the work.

6. PLOS authors have the option to publish the peer review history of their article (what does this mean? ). If published, this will include your full peer review and any attached files.

**Do you want your identity to be public for this peer review?** For information about this choice, including consent withdrawal, please see our Privacy Policy .

Reviewer #1: No

Reviewer #2: No

Reviewer #3: **Yes: ** V Rodilla

---

## [Author Response · Author response to Decision Letter 1]

23 Feb 2025

Responses to the reviewers

Referee #1 (Remarks to the Author):

The manuscript explores the potential of combining T-DM1, an antibody-drug conjugate, with autophagy inhibitors to enhance its cytotoxicity in HER2-low gastric carcinoma cells. The authors concluded that T-DM1 alone exhibits limited cytotoxicity in HER2-low carcinoma cells due to the activation of cytoprotective autophagy. However, inhibiting autophagy pharmacologically significantly increases T-DM1’s cytotoxic effects by promoting its lysosomal fusion and facilitating the release of its active cytotoxic agent, emtansine.

Comments:

The study largely repurposes previously explored concepts by combining T-DM1 with autophagy inhibitors, an approach that has already been examined in other contexts. It does not provide groundbreaking insights or novel mechanisms specific to HER2-low gastric cancer.

Thank you very much for your constructive comments. Previous studies mainly focused on HER2-positive tumors, but our study focuses on HER2-low gastric cancer initially. In fact, the proportion of HER2-low gastric cancer (21%) is close to HER2-positive gastric cancer (23.2%), and groundbreaking insights or novel mechanisms specific need to be further studied (Nakayama I, Takahari D, Chin K, et al. Incidence, clinicopathological features, and clinical outcomes of low HER2 expressed, inoperable, advanced, or recurrent gastric/gastroesophageal junction adenocarcinoma. ESMO Open. 2023 Aug;8(4):101582.).

While the manuscript mentions the involvement of the Akt/mTOR pathway and autophagy modulation, it fails to thoroughly investigate the molecular interactions or provide sufficient evidence to establish causal relationships between autophagy inhibition and the enhanced efficacy of T-DM1.

Thanks very much for your valuable suggestion. The Akt/mTOR pathway is an important pathway regulating metabolism, proliferation and survival and plays a key role in maintaining cell homeostasis in cells. This signaling pathway can participate in the occurrence and development of many diseases by mediating autophagy. As a major regulatory molecule of cell growth and metabolism, mTOR can promote anabolic processes and inhibit catabolic processes (autophagy). Activated Akt can stimulate mTOR through a series of signal transduction processes, and ultimately inhibit autophagy. Our study suggests that the Akt/mTOR pathway and its downstream molecules were closely involved in the induction of autophagy by T-DM1 in SGC7901 cells. Furthermore, autophagy inhibition can significantly enhance T-DM1-mediated cytotoxicity and apoptosis. The modified contents were as follows.

“…Together, these results suggest that the Akt/mTOR pathway is restrained in the induction of autophagy by T-DM1 in SGC7901 cells.…”

The figures and results sections include excessive repetition, particularly in the presentation of apoptotic and autophagy-related assays. Streamlining this information would improve clarity and enhance the manuscript's readability.

Thanks very much for your valuable advice. Based on your suggestion, I simplified Figure 2. I placed the statistics from the FCM and Figure 2c into the supplementary data to improve the usability of the article.

The Figure 2. are modified and replaced appropriately.

Supplementary Figure S4. are added appropriately.

The modified contents were as follows.

“…Figure 2 T-DM1 dose-dependently induced apoptosis in SGC7901 cells. (A) Apoptotic SGC7901 cells were stained with Annexin V-FITC/PI and observed after being incubated with T-DM1 for the indicated time (48 h) and were then examined by FCM. (B) Immunoblot analysis was used to measure the expression of apoptotic proteins (cleaved PARP, cleaved caspase 9, and cleaved caspase 3) after SGC7901 cells were treated with T-DM1 for 48 hours.…”

“…Supplementary Figure S4. (A) Apoptotic cells examined by FCM for the indicated time (48 h) were counted and analyzed, with results shown in bar charts (n = 3; mean ± S.D.; ***P < 0.001). (B-D) Densitometric values of the expression of apoptotic proteins (cleaved PARP, cleaved caspase 9, and cleaved caspase 3) after SGC7901 cells treated with T-DM1 for 48 hours were calculated using ImageJ software and normalized to the value of the corresponding vehicle bands. The values of the vehicle group were adjusted to 1.0 and are shown as the mean ± S.D. Student’s t-test was used to analyze the data. *P < 0.05 and ***P < 0.001.…”

The manuscript lacks detailed descriptions of the statistical analyses performed, including criteria for significance and the sample sizes used for each experiment. This omission undermines the reproducibility and robustness of the findings.

Thank you for your advice. GraphPad Prism 8 (San Diego, CA) was used to analyze the data. The results were analyzed by Student’s t-test (n = 3 independent experiments) and are presented as the mean ± S.D. Differences were believed statistically significant when P < 0.05. According to your comment, we have added related data in our revised manuscript. The modified contents were as follows.

Figure 1 T-DM1 exhibited limited cytotoxicity and was taken up by SGC7901 cells. (A) CCK-8 assays were used to evaluate the survival rates of SGC7901 cells treated with T-DM1 in a concentration-dependent manner for 72 hours (mean ± S.D.; ***P < 0.001, n = 3).

The results were analyzed by Student’s t-test (n = 3 independent experiments). The sample size and mean ± S.D. in Figure 1A are shown in the table.

Figure 3 Autophagosomes formation and autophagic flux were induced by T-DM1 in SGC7901 cells. (A) Immunoblot analysis showed the expression of autophagic proteins (SQSTM1 and LC3-Ⅱ) after SGC7901 cells were incubated with T-DM1 for 48 hours. The densitometric values were estimated by ImageJ software and normalized to the vehicle values. The data in the vehicle group were adjusted to 1.0, and data from three independent experiments are shown showed as the mean ± S.D. (Student’s t-test; *P < 0.05; ***P < 0.001, n = 3).

The results were analyzed by Student’s t-test (n = 3 independent experiments) and are presented as the mean ± S.D. Differences were believed statistically significant when P < 0.05.

Figure 4 Autophagy inhibition induced by T-DM1 enhanced the cytotoxicity of SGC7901 cells. (E) Autophagy inhibition with 3-MA or LY294002 reduced the viability of SGC7901 cells, as determined by relevant kits (mean ± S.D.; *P < 0.05 versus vehicle, n = 3). (F) The percentages of apoptotic SGC7901 cells were determined by FCM after incubation with 10 μg/ml T-DM1 and autophagy inhibitors (3-MA or LY294002). The results are presented in the side bar charts (***P < 0.001, n = 3).

Figure 5 The Akt/mTOR pathway and downstream components were involved in autophagy induced by T-DM1. (B) The densitometric values of SGC7901 cell samples were evaluated using ImageJ software and normalized to the corresponding values of the vehicle group. The values of the vehicle group were adjusted to 1.0 (mean ± S.D.; *P < 0.05; **P < 0.01 versus vehicle, n = 3).

The results were analyzed by Student’s t-test (n = 3 independent experiments) and are presented as the mean ± S.D. Differences were believed statistically significant when P < 0.05.

The results of Supplementary Material were analyzed by Student’s t-test (n = 3 independent experiments) and are presented as the mean ± S.D. Differences were believed statistically significant when P < 0.05.

The study does not adequately compare the efficacy of the proposed combination therapy with existing treatment options for HER2-low gastric cancer, limiting its clinical relevance and applicability.

Thank you for your advice. In our study, the effect of combination therapy was significantly better than that of single therapy for HER2-low gastric cancer in vitro experiments (Figure 4 E and F). Clinical relevance and applicability need to be further studied.

Phrases such as "significantly enhanced cytotoxicity" are used without providing clear quantitative benchmarks, leaving the practical significance of the findings unclear.

Thanks very much for your valuable advice. Based on your suggestion, I would be more careful to use "significantly enhanced cytotoxicity".

The discussion fails to place the findings within the broader landscape of HER2-low gastric cancer research. Conflicting evidence or alternative interpretations are not sufficiently addressed, leaving gaps in the contextual analysis.

Thank you for your suggestion. The modified contents were as follows.

“…GC, a prevalent malignant tumor, is receiving increasing attention because of its poor prognosis and high incidence [2]. HER2-targeted therapies are commonly used in clinical settings, but their application in HER2-low GC patients remains limited [4]. While HER2-targeted therapies have shown effectiveness in HER2+ GC, patients with HER2-low GC exhibit poor responses to these treatments [18, 19, 20]. Classification based on clinical features cannot fully describe biological characteristics, as nearly half of the HER2 negative BC tumors currently defined exhibit some degree of IHC expression, which has recently been renamed as HER2 low expression [21]. T-DM1, which is the first antibody-drug conjugate (ADC) approved for clinical use, is designed to decompose in lysosomes and disrupt microtubule organization, thereby selectively targeting HER2+ malignant tumor cells [20, 22, 23 , 24]. However, the effectiveness of T-DM1 is largely reduced in HER2-low tumors, so T-DM1 cannot be applied in clinical practice in HER2-low tumors. In recent years, people began to gradually pay attention to HER2-low gastric cancer, hoping to improve the effectiveness of targeted therapy [25]. This limited efficacy underscores the urgent need to better understand the mechanisms by which T-DM1 can be optimized for treating HER2-low GC.…”

“…ADCs represent a promising class of chemotherapy drugs that are engineered to deliver cytotoxic agents directly to target cells via monoclonal antibodies [36]. Ideally, target cells should remain highly sensitive to ADCs. However, our study showed that T-DM1 did not exhibit a superior therapeutic advantage against HER2-low GC cells [37]. Novel ADC drugs have been tested in clinical trials in HER2-low tumors, which suggests that fewer available HER2 receptors on cancer cells may be sufficient to achieve clinical benefits [38]. We observed the processing of T-DM1 from receptor recognition to intracellular lysosomes in SGC7901 cells, finding that fusion with lysosomes was strengthened by autophagy inhibition. Increased fusion of T-DM1 with lysosomes effectively increased the cytotoxic effect of T-DM1 on SGC7901 cells. Importantly, our findings, along with previous studies, suggest that autophagy interference in the dissociation of ADCs may be a universal phenomenon across different tumors [15, 39]. Therefore, regulating autophagy could potentially alter the sensitivity of ADC-based cancer treatment and could be a viable strategy to broaden the therapeutic range of ADCs.…”

The authors assert that autophagy inhibition may have universal applications across various tumor types but provide no supporting evidence beyond the context of HER2-low gastric cancer cells. This weakens the broader impact of the conclusions.

Thanks very much for your valuable suggestion. In a previous study, we found that autophagy occurs extensively during the treatment of tumors. The study of Zhang J, et al. showed that targeting the autophagy promoted antitumor effect of T-DM1 on HER2-positive Gastric Cancer (Zhang J, et al. Targeting the autophagy promoted antitumor effect of T-DM1 on HER2-positive gastric cancer. Cell Death Dis. 2021 Mar 17;12(4):288. doi: 10.1038/s41419-020-03349-1.). Another paper by Shen, W. et al. also revealed the same antitumor effect in non-small cell lung cancer (Shen, W. et al. A novel and promising therapeutic approach for NSCLC: recombinant human arginase alone or combined with autophagy inhibitor. Cell Death Dis. 2017 Mar 30;8(3): e2720. doi: 10.1038/cddis.2017.137.). We will add the corresponding references to reinforce the conclusion of the article. The modified contents were as follows.

“…Importantly, our findings, along with previous studies, suggest that autophagy interference in the dissociation of ADCs may be a universal phenomenon across different tumors [15, 36]. Therefore, regulating autophagy could potentially alter the sensitivity of ADC-based cancer treatment. To date, it remains unclear whether modulating autophagy could be a viable strategy to broaden the therapeutic range of ADCs. …”

Referee #2 (Remarks to the Author):

In the manuscript entitled: “Enhanced cytotoxicity of T-MD1 in HER2-low carcinomas via autophagy inhibition”, Zhang et al. showed that T-DM1 induces autophagy and combination of T-DM1 with autophagy inhibitors increases T-DM1 efficacy in HER2-low gastric cancer. A potential expansion of the use of T-DM1 and may be other HER2-targeted therapies to tumors with low HER2 expression is significant since the HER2-low patient subpopulation usually exhibits an aggressive disease progression. Despite that, there are several major points that need to be addressed as listed below:

Major points:

1- All the IF images need quantification.

Thanks very much for your valuable suggestion. All the IF images had been quantified. This can be seen in Supplementary Figure S1, Supplementary Figure S2. (B-D) and Supplementary Figure S5. The modified contents were as follows.

“…The brightness values of HER2 red dye staining were estimated using ImageJ software and the densitometric values were normalized to the corresponding values of the vehicle group. The values of the vehicle group were set as 1.0 (n = 3 independent experiments; mean ± S.D.; **P < 0.01 and ***P < 0.001). …”

“…(C) The brightness values of the Alexa Fluor™ 647 were evaluated using ImageJ software, and the values were normalized to the corresponding values of the vehicle group. (D) …”

“…Supplementary Figure S5. The brightness values of the Alexa Fluor™ 647 visualized by red fluorescence, Cyto-ID visualized by fluorescence and LysoTracker exhibited yellow fluorescence were evaluated using ImageJ software, and the values were normalized to the corresponding values of the vehicle group. …”

2- In Fig. 1B, it was claimed that endocytosis happens at 6 hrs, however, it is not visible from the image provided.

Thank you very much for your constructive comments. we claimed that endocytosis happening at 6 hrs was based on the uniform distribution of the labeling T-DM1 with red fluorescence on the cells. This conclusion really lacks more accurate experimental support, so we have made corresponding modifications to the paper. The modified contents were as follows.

“…Laser confocal microscopy was used to track the uptake of T-DM1 at different time points, revealing that T-DM1 bound to the HER2 receptor at 1 hour and internalized at 12 hours post-treatment (Figure 1B). …”

3- Is there any change in T-DM1 internalization upon combination of T-DM1 with autophagy inhibitors.

Thank you for your asking. We speculated that T-DM1 internalization upon combination of T-DM1 with autophagy inhibitors should be enhanced, thus enhancing cytotoxicity (Figure 6 and Supplementary Figure S5).

4- It is not clear how the apoptotic cells were quantified. Were they the Annexin V+ and PI- cells corresponding to early apoptosis or Annexin V+ PI+ cells corresponding to late apoptosis?

Thank you very much for your constructive comments. We quantitatively counted early apoptosis (the Annexin V+ and PI- cells). The modified contents were as follows.

“…(F) The percentages of early apoptotic SGC7901 cells (the Annexin V+ and PI- cells) were determined by FCM after incubation with 10 μg/ml T-DM1 and autophagy inhibitors (3-MA or LY294002). The results are presented in the side bar charts (***P < 0.001, n = 3).…”

“…Apoptotic cells (the Annexin V+ and PI- cells) examined by FCM for the indicated time (48 h) were counted and analyzed, with results shown in bar charts (n = 3; mean ± S.D.; ***P < 0.001).…”

5- Is DM1 treatment also inducing autophagy induction upon mTOR inhibition?

Thank you for your asking. Our current experimental results can only prove than the Akt/mTOR pathway is restrained in the induction of auto

---

## [Decision Letter · Decision Letter 1]

16 Mar 2025

Enhanced cytotoxicity of T-DM1 in HER2-low carcinomas via autophagy inhibition

PONE-D-24-51611R1

Dear Dr. Xin,

We’re pleased to inform you that your manuscript has been judged scientifically suitable for publication and will be formally accepted for publication once it meets all outstanding technical requirements.

Kind regards,

Ruo Wang

Academic Editor

PLOS ONE

Additional Editor Comments (optional):

Reviewers' comments:

Reviewer's Responses to Questions

**Comments to the Author**

1. If the authors have adequately addressed your comments raised in a previous round of review and you feel that this manuscript is now acceptable for publication, you may indicate that here to bypass the “Comments to the Author” section, enter your conflict of interest statement in the “Confidential to Editor” section, and submit your "Accept" recommendation.

Reviewer #1: All comments have been addressed

2. Is the manuscript technically sound, and do the data support the conclusions?

Reviewer #1: Yes

3. Has the statistical analysis been performed appropriately and rigorously? 

Reviewer #1: Yes

4. Have the authors made all data underlying the findings in their manuscript fully available?

Reviewer #1: (No Response)

5. Is the manuscript presented in an intelligible fashion and written in standard English?

Reviewer #1: (No Response)

6. Review Comments to the Author

Reviewer #1: The authors answers all comments and they improve the manuscript. The manuscript is accepted for publication

7. PLOS authors have the option to publish the peer review history of their article (what does this mean? ). If published, this will include your full peer review and any attached files.

**Do you want your identity to be public for this peer review?** For information about this choice, including consent withdrawal, please see our Privacy Policy .

Reviewer #1: No

---

## [Editor Report · Acceptance letter]

PONE-D-24-51611R1

PLOS ONE

Dear Dr. Xin,

I'm pleased to inform you that your manuscript has been deemed suitable for publication in PLOS ONE. Congratulations! Your manuscript is now being handed over to our production team.

Kind regards,

on behalf of

Dr. Ruo Wang

Academic Editor

PLOS ONE